# Segmental Advantage Estimation: Enhancing PPO for Long-Context LLM Training

## Abstract

Training Large Language Models (LLMs) for reasoning tasks is increasingly driven by Reinforcement Learning with Verifiable Rewards (RLVR), where Proximal Policy Optimization (PPO) provides a principled framework for stable policy updates. However, the practical application of PPO is hindered by unreliable advantage estimation in the sparse-reward RLVR regime. This issue arises because the sparse rewards in RLVR lead to inaccurate intermediate value predictions, which in turn introduce significant bias when aggregated at every token by Generalized Advantage Estimation (GAE). To address this, we introduce Segmental Advantage Estimation (SAE), which mitigates the bias that GAE can incur in RLVR. Our key insight is that aggregating $n$-step advantages at every token(as in GAE) is unnecessary and often introduces excessive bias, since individual tokens carry minimal information. Instead, SAE first partitions the generated sequence into coherent sub-segments using low-probability tokens as heuristic boundaries. It then selectively computes variance-reduced advantage estimates only from these information-rich segment transitions, effectively filtering out noise from intermediate tokens. Our experiments demonstrate that SAE achieves superior performance, with marked improvements in final scores, training stability, and sample efficiency. These gains are shown to be consistent across multiple model sizes, and a correlation analysis confirms that our proposed advantage estimator achieves a higher correlation with an approximate ground-truth advantage, justifying its superior performance.

## 1 Introduction

Recent advances in large language model (LLM) training for reasoning tasks have been increasingly driven by Reinforcement Learning with Verifiable Rewards (RLVR), which leverages automated verifiers to provide binary feedback on objectively measurable tasks. In this context, Group Relative Policy Optimization (GRPO) (Shao et al., 2024) has been favored for reducing system complexity and easing implementation. However, this simplicity sacrifices fine-grained credit assignment over individual reasoning steps. In contrast, Proximal Policy Optimization (PPO) provides a more principled advantage estimation framework that can supply fine-grained training signals (Hu et al., 2025; Yue et al., 2025) via a learned critic model.

Despite its theoretical promise, the practical application of PPO in RLVR is hindered by unreliable advantage estimation. As noted by Kazemnejad et al. (2025), this difficulty stems from the sparse rewards inherent to the RLVR regime, which render intermediate value predictions highly unreliable. This creates difficulties for Generalized Advantage Estimation (GAE) (Schulman et al., 2016) used in PPO, which aggregates exponentially discounted mixtures of per-token n-step advantages. Specifically, GAE depends on value predictions at every token position; and in RLVR where these predictions are inherently unreliable, this fine-grained token-level bootstrapping process may introduce more estimation bias than useful training signal. Consequently, such bias-amplifying token-wise advantage aggregation mechanism of GAE may compromise the quality of policy updates in long-horizon reasoning tasks.

Existing attempts to solve the above challenges present certain limitations. One common solution (Hu et al., 2025) involves setting $\lambda = 1$ to obtain unbiased Monte Carlo estimates, but this sacrifices both GAE's core advantage of variance reduction and its potential for effective credit assign-

ment, resulting in unstable gradient estimates and suboptimal learning dynamics. Another approach (Yue et al., 2025) increases $\lambda$ values for longer sequences to better handle heterogeneous response lengths, yet this requires extensive hyperparameter tuning and still suffers from the inaccurate value estimates. Although these adaptations report empirical gains, they operate within GAE's framework and leave intact GAE's bias-amplifying token-level advantage aggregation, limiting the quality of estimated advantages.

To mitigate the excessive bias introduced by GAE, our key insight is that the majority of token-level advantage bootstrapping operations are unnecessary and can even be harmful. Individual tokens often carry minimal information and contribute little effective guiding signal, yet their inclusion in the bootstrapping process introduces substantial estimation bias. This suggests an intuitive solution: strategically selecting a sparse subset of representative $n$-step advantages for estimation, rather than indiscriminately aggregating them at every token position. To achieve this, we notice that the content generated by LLMs can be organized into semantically coherent segments, such as mathematical sub-derivations or logical subclaims. This structure provides a natural mechanism for our approach: instead of applying GAE at the token level, we restrict advantage computation to the semantic transition boundaries where the underlying reasoning advances. By doing so, we significantly reduce the number of low-information bootstrap points, thereby lowering bootstrapping bias and providing a more stable foundation for PPO-style policy optimization in RLVR.

Building on these insights, we introduce Segmental Advantage Estimation (SAE) to address the challenges of long-sequence advantage estimation in RLVR reasoning tasks. We first propose a heuristic method to partition the entire response sequence into semantically coherent segments, which identifies low-probability tokens as segment boundaries. This heuristic is motivated by the observation that within a continuous reasoning segment, token generation is highly predictable (i.e. with high probability); conversely, transitions between distinct reasoning steps force the model to navigate higher uncertainty, which naturally manifests as the generation of high-surprisal, low-probability tokens that serve as effective semantic boundaries. When aggregating the mixture of n-step advantages to obtain a variance-reduced advantage estimate, our method selectively includes only the estimates originating from segment boundaries instead of every token as GAE does. This design effectively filters out the noisy value predictions from intermediate tokens within each segment, ensuring that the final advantage estimation is built upon a foundation of more reliable and informative states.

We present an extensive empirical evaluation of our proposed method within the domain of mathematical problem-solving. Our approach is benchmarked against several strong baselines across four out-of-distribution test sets (AIME'24, AIME'25, AMC, and BeyondAIME). The experimental results demonstrate that our method achieves superior performance, exhibiting marked improvements in final scores, training stability, and sample efficiency over other baselines. An ablation study on model size further establishes that these gains are consistent across 4B/8B/14B models, demonstrating the broad applicability of our method. Finally, we provide a direct justification for our method's effectiveness through a correlation analysis, which reveals that our proposed advantage estimator achieves the highest correlation with an approximate ground-truth advantage, thereby explaining its superior performance.

## 2 RELATED WORKS

**Reinforcement Learning with Verifiable Rewards**    Recent work on reinforcement learning with verifiable rewards (RLVR) has yielded substantial gains in LLM reasoning, especially in mathematics and programming where correctness provides reliable signals. Some works Chen et al. (2024); Feng et al. (2024) utilize searching method such as MCTS to construct examples with positive rewards which are then fed to the LLMs to enhance its ability. Recent breakthroughs, exemplified by DeepSeek R1 (DeepSeek-AI et al., 2025), have proven that the exclusive use of reinforcement learning (RL) can considerably push the ceiling of reasoning abilities, a development that has greatly catalyzed the recent enhancement of reasoning in LLMs. R1 (DeepSeek-AI et al., 2025) also introduces a "zero RL" paradigm that elicits reasoning directly from a base model without additional supervised finetuning. The success of R1 has motivated many variants of GRPO: DAPO (Yu et al., 2025) surfaces failure modes such as entropy collapse and proposes four mitigations; REINFORCE++ (Hu, 2025) employs unbiased global advantage normalization to improve stabil-

ity; GSPO (Zheng et al., 2025) replaces token-wise clipped importance ratios with sequence-level clipping, improving MoE training stability; and LitePPO (Liu et al., 2025) demonstrates that a minimalist combination of two techniques unlock the learning capability of models with vanilla PPO loss. Despite the simplicity of value-model-free approaches (GRPO and its variants), recent studies report that PPO can achieve superior performance (Yue et al., 2025; Hu et al., 2025), likely due to its more granular credit assignment. Value-model-based Approach therefore remains a promising direction with greater upside for RLVR tasks.

**Value-Model-Based Approaches in RLVR** Recent value-model-based RLVR work for long-horizon chain-of-thought reasoning focuses on robust advantage estimation, mitigation of value-model bias, and effective credit assignment. Open-Reasoner-Zero (Hu et al., 2025) demonstrates that a minimalist setup—vanilla PPO with GAE($\lambda$=1, $\gamma$=1), simple rule-based rewards, and no KL regularization—scales performance and response length. VAPO (Yue et al., 2025) increases $\lambda$ values of GAE for longer sequences to better handle heterogeneous response lengths and reduce the bias brought by the value model. VC-PPO (Yuan et al., 2025) addresses value initialization bias and reward signal decay by pretraining the value function and decoupling GAE computation between the actor and critic. T-PPO (Fan et al., 2025) introduces Extended GAE to compute advantages from incomplete responses while maintaining policy stability. Although these methods have been shown to be effective in their respective domains, they typically inherit the token-level advantage bootstrapping of standard GAE and are therefore orthogonal to our approach.

## 3 PRELIMINARY

### 3.1 PROXIMAL POLICY OPTIMIZATION (PPO)

Proximal Policy Optimization (PPO) is a state-of-the-art policy gradient method in reinforcement learning developed by Schulman et al. (2017), whose surrogate optimization objective can be formatted as:

$$\mathcal{L}^{PPO}(\theta) = \mathbb{E}_{x \sim \mathcal{D}} \left[ \sum_{t=1}^{T} \min \left( r_t(\theta) A_t^{\text{GAE}}, \text{clip}(r_t(\theta), 1 - \epsilon, 1 + \epsilon) \hat{A}_t \right) \right] \quad (1)$$

Here, $T$ is the sequence length, $A_t^{\text{GAE}}$ is the GAE estimator, $r_t(\theta) = \frac{\pi_\theta(a_t|s_t)}{\pi_{\text{old}}(a_t|s_t)}$ denotes the importance ratio, which quantifies the policy change probability for action $a_t$ in state $s_t$ between the current policy $\pi_\theta$ and the behavior policy $\pi_{\text{old}}$.

### 3.2 PROBLEM SETTING FOR RLVR

Since we are mainly interested in LLM reasoning tasks with outcome rewards, therefore in this paper we assume the reward is assigned to the final token of generated response:

$$r_t = \begin{cases} 0, & t < T \\ \mathbb{I}[\text{correct}], & t = T \end{cases}. \quad (2)$$

Since the reward is particularly sparse, we also set the reward discount factor $\gamma = 1$ in the remaining paper, which is the common practice in LLM community (Yue et al., 2025; Hu et al., 2025). Under these conditions, the temporal difference error becomes:

$$\delta_t = r_t + V(s_{t+1}) - V(s_t) = \begin{cases} V(s_{t+1}) - V(s_t), & t < T - 1 \\ \mathbb{I}[\text{correct}] - V(s_{T-1}), & t = T - 1 \end{cases} \quad (3)$$

For simplicity, we assume $V(S_T) = \mathbb{I}[\text{correct}]$. GAE is defined as the exponentially-weighted average of $l$-step advantage estimators $A_t^{(l)} := V(S_{t+l}) - V(s_t)$, and can be simplified to the weighted sum of $\delta_t$ (defined in Eq.3):

$$A_t^{\text{GAE}}(\lambda) := (1 - \lambda) \left( A_t^{(1)} + \lambda A_t^{(2)} + \lambda^2 A_t^{(3)} + \dots \right) = \sum_{l=0}^{T-t} \lambda^l \delta_{t+l}. \quad (4)$$

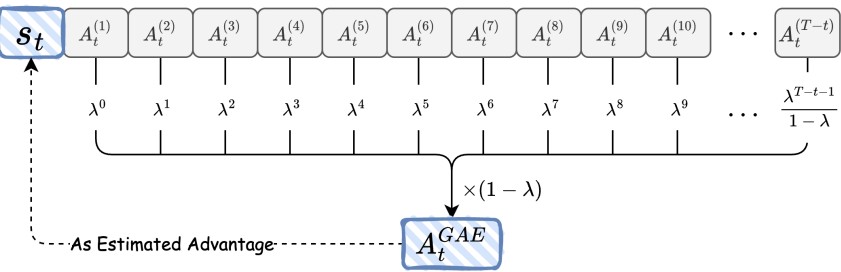

(a) GAE: token-level advantage boostrapping.

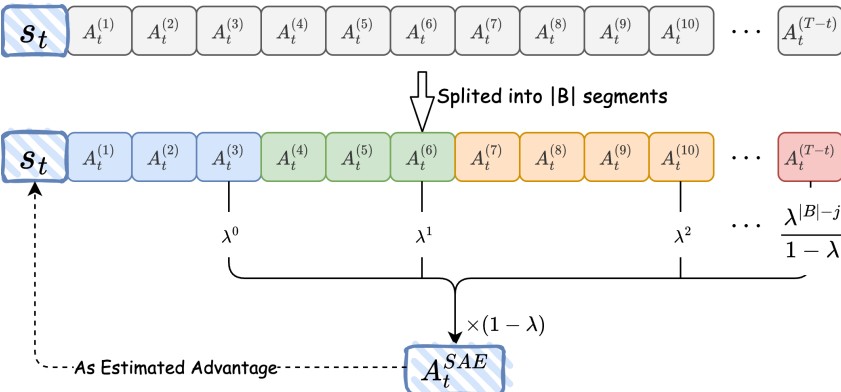

(b) SAE: Segment-level Advantage Boostrapping.

Figure 1: The sketch of GAE and SAE. Instead of bootstrapping at every token as GAE does, SAE first partitions the sequence into semantically coherent segments, and then computes advantage estimators only at the boundaries of these segments.

## 4 METHODS

In order to provide better advantage estimation for RLVR tasks, we introduce Segmental Advantage Estimation (SAE). The difference of GAE and SAE is illustrated in Figure 1. Instead of bootstrapping at every token as GAE does, SAE first partitions the sequence into a small number of semantically coherent segments, and then computes advantage estimators only at the boundaries of these segments. This design directly confronts the bias amplification problem by drastically reducing the number of noisy, value-based estimators in the GAE sum. By anchoring the advantage calculation to a sparse set of key states, SAE produces a more stable and reliable training signal.

In this section, we first describe our probability-based segmentation method and the segment-level advantage estimation in Section 4.1. We further provide a theoretical analysis on the benefits of SAE's segment-level boostrapping in Section 4.2.

### 4.1 SEGMENTAL ADVANTAGE ESTIMATION

#### 4.1.1 PROBABILITY-BASED SEGMENTATION

The efficacy of SAE is critically contingent upon its strategy for sequence segmentation. A well-chosen set of segmentation points, marking significant semantic shifts, is essential for isolating a high-quality advantage signal. To ensure the generalizability of our approach, we deliberately avoid task-specific heuristics, such as keywords matching (*wait, step* or \n). Instead, we propose a method that partitions the sequence by leveraging intrinsic features of the model's own output tokens.

Our core intuition is that the model's confidence in its own predictions serves as a powerful indicator of semantic structure. When the model generates a sequence of tokens that are highly probable given

the context (e.g., completing a common phrase), it is operating within a coherent semantic segment. Conversely, a token generated with very low probability represents a point of high "surprise" for the model, which potentially marks a meaningful transition in the generated text, such as the introduction of a new argument or a shift in reasoning. These are precisely the semantic boundaries we wish to identify.

Building on the insight above, we define a content-dependent binary segmentation function $f_s :$ $\{1, 2, \ldots, T\} \to \{0, 1\}$ for a given response sequence $s = (s_1, s_2, \ldots, s_T)$, which identifies semantic boundaries as tokens with generation probability below threshold $p$:

$$f_s(t) = \begin{cases} 1 & \text{if } P_{\text{model}}(s_t | s_{<t}) < p, \\ 0 & \text{otherwise,} \end{cases} \tag{5}$$

where $p$ is a threshold hyperparameter that controls the granularity of the segmentation.

### 4.1.2 ADVANTAGE ESTIMATION

**Selective Advantage Integration**    Rather than computing advantages at every token position, SAE selectively incorporates advantages computed at segment boundaries, as shown in Figure 1b. Let $\mathcal{B} = \{t : f_s(t) = 1\} \cup \{T\}$ denote the set of segment boundary positions, including the terminal position $T$. We represent this set as an ordered sequence $\mathcal{B} = \{t_k\}_{k=1}^{|\mathcal{B}|}$, where $t_k < t_{k+1}$ for all $k$.

The core insight is to construct a GAE-style weighted combination using only these strategically selected advantage estimates:

$$A_t^{\text{SAE}} = (1 - \lambda) \sum_{k : t_k > t, t_k \in \mathcal{B}, t_k < T} \lambda^{k-j} A_t^{(t_k - t)} + \lambda^{|B| - j} A_t^{(T-t)}, \tag{6}$$

where $j := \min\{k : t_k > t, t_k \in \mathcal{B}\}$ is the index such that $t_j$ is the segment boundary immediately after position $t$, and $A_t^{(t_k - t)} := V(s_{t_k}) - V(s_t)$ is the $(t_k - t)$-step boostraping advantage estimate.

**Reformulation via Temporal Difference Errors**    To derive a computationally efficient form, we express the selected advantage estimates in terms of temporal difference errors. For any segment boundary $t_k \in \mathcal{B}$, the $(t_k - t)$-step advantage can be written as: $A_t^{(t_k - t)} = \sum_{i=0}^{t_k - t - 1} \delta_{t+i}$. Substituting this into our SAE formulation and rearranging terms, we obtain:

$$A_t^{\text{SAE}} = (1 - \lambda) \sum_{k : t_k > t, t_k < T} \lambda^{k-j} \sum_{i=0}^{t_k - t - 1} \delta_{t+i} + \lambda^{|\mathcal{B}| - j} \sum_{i=0}^{T-t-1} \delta_{t+i}. \tag{7}$$

By changing the order of summation, this can be simplified to:

$$A_t^{\text{SAE}} = \sum_{l=0}^{T-t-1} \left( \prod_{i=0}^{l-1} \lambda_{\text{SAE}}(t+i) \right) \delta_{t+l}, \tag{8}$$

where $\lambda_{\text{SAE}}(t)$ is a adaptive decay parameter, which implements segment-aware weighting:

$$\lambda_{\text{SAE}}(t) = \begin{cases} 1 & \text{if } f_s(t+1) = 0 \text{ (intra-segment)}, \\ \lambda & \text{if } f_s(t+1) = 1 \text{ (cross-segment)}. \end{cases} \tag{9}$$

The detailed proof from Eq.7 to Eq.8 is provided as TheoremA.1 in the Appendix. Actually, Eq.8 is equivalent to a GAE-like formulation where the discount factor is adaptively applied only at segment boundaries: temporal difference errors within the same semantic segment receive equal weighting (no decay), while errors across segment boundaries are subject to exponential decay $\lambda$.

**Recursive Formulation**    Finally, the SAE advantage admits a computationally efficient recursive formulation:

$$A_t^{\text{SAE}} = \delta_t + \lambda_{\text{SAE}}(t) \cdot A_{t+1}^{\text{SAE}}, \tag{10}$$

where $\lambda_{\text{SAE}}(t)$ is defined in Eq.9.

The recursive form Eq.10 reveals that SAE retains the computational elegance of GAE. It can be seamlessly integrated into existing PPO implementations by simply making the decay factor conditional, as shown in Eq. 9. Thus, our method provides a more robust advantage signal without incurring significant computational overhead, offering a powerful yet practical enhancement.

## 4.2 THEORETICAL ANALYSIS

In the previous sections, we introduced SAE with the core intuition that replacing token-level bootstrapping with segment-level bootstrapping can mitigate the propagation of noise from the learned value function. To formally ground this intuition, we now provide a theoretical analysis that quantifies the relationship between segmentation and bias. Our goal is to demonstrate that by reducing the frequency of bootstrapping through segmentation, SAE can achieve a tighter bias bound compared to traditional token-level GAE.

While our practical method employs a dynamic, probability-based segmentation, for the purpose of a tractable theoretical proof, we analyze the case of a uniform segmentation strategy. This simplification allows us to derive a clear, interpretable relationship between the average segment length ($M$) and an upper bound on the estimation bias. The following theorem presents our theoretical result:

**Theorem 4.1.** *(Upper bound of bias for* SAE *under uniform segmentation) Consider the SAE advantage estimation at step $t = 0$ as defined in Eq. 8:*

$$A_0^{SAE} = \sum_{l=0}^{T} \left( \prod_{i=0}^{l-1} \lambda_{SAE}(i) \right) \delta_l, \tag{11}$$

*where $\lambda_{SAE}(t)$ is given by Eq. 9.*

*We make the following assumptions:*

1. *(Uniform segmentation) To facilitate a tractable theoretical analysis, we assume a simplified, uniform segmentation model defined as:*

$$f_s(t) = \begin{cases} 1 & \text{if } t \equiv 0 \pmod{M} \\ 0 & \text{otherwise}. \end{cases} \tag{12}$$

   *This function places boundaries at regular intervals, creating segments of a fixed length $M$. We also assume the segment length $M$ divides the horizon $T$ exactly, i.e., $T = nM$ for some integer $n$.*

2. *(Value function approximation): $V(s_t) = V^*(s_t) + \varepsilon(s_t)$ where $V^*(s_t)$ is the ground-truth value function and $\varepsilon(s_t)$ is the error term. we also assume that $|\varepsilon(s_t)| \leq \alpha \exp\left(\frac{T-t}{\beta}\right)$ with $\alpha > 0, \beta > 0$, which models a common scenario where the value function's approximation error grows for states further away from the end of the trajectory.*

*Then the bias of SAE satisfies:*

$$\left| bias_{A_0^{SAE}} \right| \leq \alpha \exp\left(\frac{T}{\beta}\right) \left[ 1 + \frac{1-\lambda}{\exp\left(\frac{M}{\beta}\right) - \lambda} \right]. \tag{13}$$

*Consequently, the bias upper bound is inversely related to M: larger values of M yield tighter bounds.*

The detailed proof is provided in the Appendix B. The above theorem demonstrates that the bias decreases as the segment length $M$ increases. Consequently, for a given discount parameter $\lambda$, SAE may achieve a tighter bias bound compared to conventional token-level GAE, which corresponds to the special case where $M = 1$. This reveals that segmentation-based advantage estimation provides an alternative mechanism for bias control beyond the traditional approach of tuning the discount parameter $\lambda$, offering practitioners an additional mechanism to regulate bias in advantage estimation.

## 5 EXPERIMENTS

### 5.1 SETUP

**Models and Datasets** We employ the **Qwen3-8B-base** model as our policy backbone and focus on the domain of mathematical problem solving. For our RL training, we utilize the DAPO-Math-17k dataset (Yu et al., 2025), a curated collection of 17,000 high-quality mathematical problems.

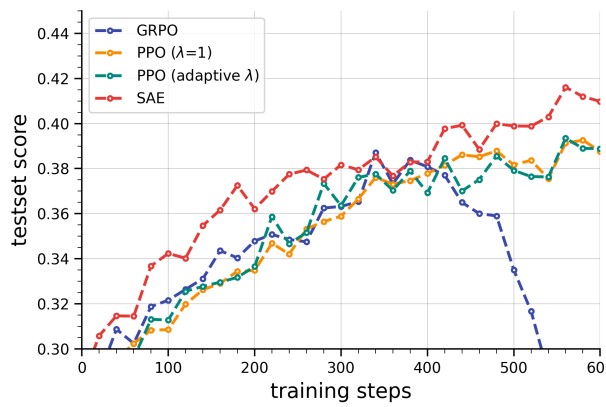

Figure 2: Macro-averaged test-set scores across four test sets for all methods training on the Qwen3-8B-base. SAE maintains consistent improvements against baselines throughout the training process.

Table 1: Benchmark results for all methods trained on Qwen3-8B-base. SAE achieves highest average score with a margin of 2.09 percentage points over the strongest baseline.

| Benchmark | GRPO* | PPO | | |
| --- | --- | --- | --- | --- |
| | | $\lambda = 1$ | adaptive $\lambda$ | SAE(ours) |
| AIME'24 | 35.42 | 34.79 | 35.42 | **38.54** |
| AIME'25 | 27.50 | 29.17 | 25.42 | **30.21** |
| AMC | 74.10 | 74.85 | **78.39** | 77.56 |
| BeyondAIME | 15.31 | 16.25 | 16.33 | **17.62** |
| **Average** | 38.08 | 38.76 | 38.89 | **40.98** |

Note: GRPO* is evaluated at 400 steps due to training instability(see Figure 2); all other methods are evaluated at 600 steps.

To evaluate out-of-distribution generalization and performance on problems of varying difficulty, we assess the model's zero-shot performance on four held-out test sets: AIME24, AIME25, AMC, and BeyondAIME. These benchmarks collectively probe the model's ability to generalize across different annual versions of the AIME competition (AIME24/25), to problems of moderate difficulty (AMC), and to a broader and more challenging distribution of reasoning tasks (BeyondAIME).

**Implementation Details**  During each training step, we sample 4096 rollouts from 512 prompts, generating 8 rollouts per prompt. The rollouts are generated with a temperature of 0.6 and a maximum response length of 8192 tokens. The actor and value models are updated with a batch size of 4096, using learning rates of $1 \times 10^{-6}$ and $1 \times 10^{-5}$, respectively. Following (Yue et al., 2025), we pre-train the value networks to provide a solid foundation for all PPO-based methods. We do not apply a KL or entropy loss, updating the actor solely with the PPO loss (see Eq.1). For SAE, we set $p = 0.2$ in Eq.5 universally for all experiments without further hyperparameter tuning.

## 5.2 MAIN RESULTS

**Baselines**  To comprehensively evaluate the efficacy of our proposed method, we benchmark it against several representative baselines. All baselines were configured with hyperparameters identical to those of our proposed method to ensure a fair comparison. The selected baselines are as follows: (1) **GRPO** (Shao et al., 2024): This approach does not utilize a value function to aid in the token-level advantage estimation. (2) **PPO** ($\lambda = 1$): A common configuration in RLVR where the parameter $\lambda$ is set to 1 (Hu et al., 2025). (3) **PPO (adaptive $\lambda$)**: This method employs the adaptive lambda strategy from VAPO(Yue et al., 2025), where $\lambda$ is a function of the generation length $l$, defined as $\lambda = 1 - \frac{1}{0.2l}$.

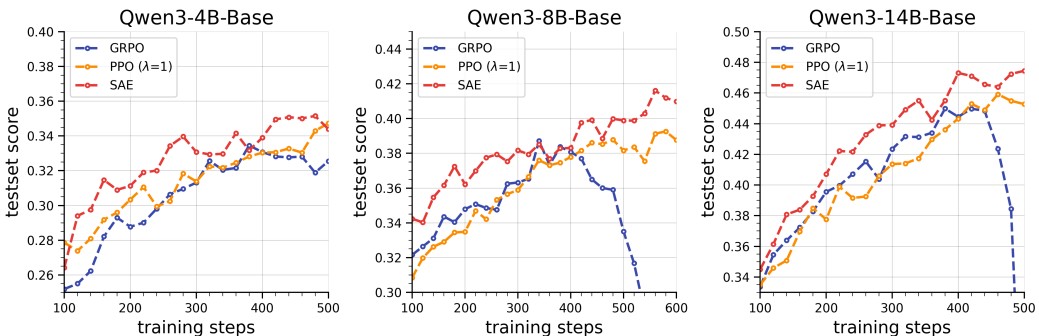

Figure 3: Results of different methods across model sizes (4B/8B/14B). SAE consistently outperforms other baselines.

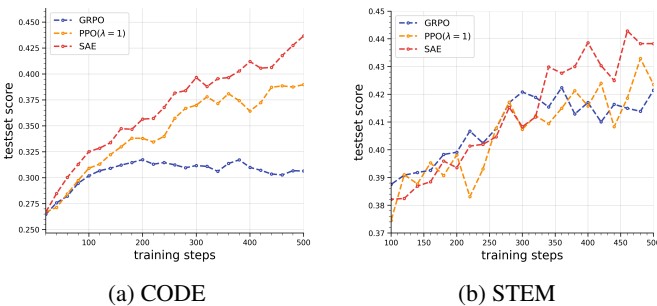

(a) CODE

(b) STEM

Figure 4: Results of additional reasoning domains (CODE and STEM). SAE consistently outperforms other baselines.

**Results** As shown in Table 1, our approach achieves the highest average score, with a margin of 2.09 percentage points over the strongest baseline. The training dynamics in Figure 2 further corroborate our approach's effectiveness, with SAE exhibiting superior sample efficiency from early training stages and maintaining consistent improvements throughout the entire process. Results in Table 1 Figure 2 demonstrate that our proposed method demonstrate superior performance across all evaluation benchmarks and exhibit enhanced training stability compared to existing baselines.

Notably, while GRPO suffers from training instability and performance collapse after approximately 400 training steps, all PPO-based variants including our proposed methods maintain stable convergence without deterioration. This observation aligns with findings in (Hu et al., 2025), suggesting that PPO-based optimization may provide inherently better stability properties compared to GRPO for RLVR tasks.

## 5.3 ABLATION STUDY

### 5.3.1 CONSISTENT GAINS ACROSS MODEL SIZES AND DOMAINS

To examine the generality of the proposed SAE approach, we conducted experiments to assess its robustness across two key dimensions: model scale and application domain. Our goal is to demonstrate that the benefits of SAE are not confined to a specific model capacity or a narrow task, but rather represent a fundamental improvement in the advantage estimation process. Owing to the computational cost of full-scale training, we restricted the comparison to two baselines: GRPO and PPO (with $\lambda = 1$). All methods were trained under the same setting as in Section 5.1, and other training details (e.g. model/train set/test set/...) are described in Appendix D.

First, we evaluated performance across three base models of different sizes: Qwen3-4B, Qwen3-8B, and Qwen3-14B, focusing on the mathematical reasoning domain. The performance curves are summarized in Figure 7. Across all three parameter scales, SAE achieves higher performance than both baselines. The consistent relative advantage observed at 4B, 8B, and 14B suggests that the

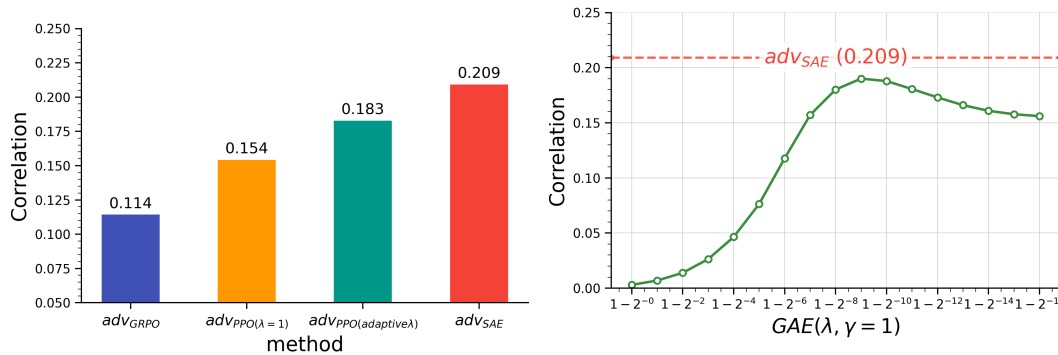

(a) Correlation of all estimators with $A^*$

(b) Correlation of GAE($\lambda$) with $A^*$, as a function of $\lambda$

Figure 5: Correlation of different methods with approximate ground-truth advantage $A^*$, which is obtained by extensive Monte Carlo sampling. SAE and $A^*$ are the most highly correlated among all methods.

improvements brought by SAE are not confined to a particular capacity regime and remain stable as the underlying model size increases. We further observe that GRPO exhibits a degradation in performance after approximately 400 training steps across all settings, whereas PPO-based methods do not, providing additional evidence of the superior stability of PPO.

Second, to validate the effectiveness of SAE beyond mathematical reasoning, we extended our evaluation to two additional challenging domains: code generation and general STEM problems. As shown in Figure 3, SAE consistently outperform other baselines in both domains. Notably, in the code domain we observed that while GRPO's performance on the test set began to stagnate after approximately 200 steps. In contrast, SAE's test performance continued to climb. This suggests that SAE, by providing more fine-grained credit assignment, can potentially unlock a higher level of performance and better generalization.

### 5.3.2 Correlation with an Approximate Ground-Truth Advantage

To directly assess the relative quality of different advantage estimators, we construct an approximate ground-truth advantage and evaluate each method by its correlation with this reference. Specifically, we randomly sample multiple trajectory segments and, for each segment, identify its initial state $s_t$ and terminal state $s_{t+m}$. Starting from each of $s_t$ and $s_{t+m}$, we perform 32 independent rollouts to obtain Monte Carlo estimates of their state values, denoted $V^*(s_t)$ and $V^*(s_{t+m})$. We then define the segment-level approximate ground-truth advantage as $A^* = V^*(s_{t+m}) - V^*(s_t)$, which we assign to every token within the segment. For every token, we also compute the corresponding advantage estimates produced by the competing methods (including SAE and other baselines), and we report the Pearson correlation between each estimator and $A^*$.

Figure 5a shows that among all evaluated methods, our SAE estimator achieves the highest correlation with $A^*$. To further investigate the potential upper bound on GAE's accuracy, we analyze how its correlation with $A^*$ changes as a function of $\lambda$ (Figure 6b). Across the entire tested range of $\lambda$, the correlation of GAE with $A^*$ remains uniformly below that of SAE. Taken together, these experiments provide direct evidence that SAE yields a more reliable and precise approximation of the true advantage. This improved estimation accuracy helps explain why SAE more effectively guides actor updates, as demonstrated in previous sections.

### 5.3.3 Robustness to the Threshold of Probability-based Segmentation

The segmentation threshold $p$ (Eq 5) is a key hyperparameter in SAE, as it directly controls the granularity of the sequence partitioning. A crucial question is how sensitive SAE's performance is to this parameter. Therefore, in this section we conducted an ablation study to analyze this sensitivity.

We conducted an ablation study across a range of values ($p \in \{0.05, 0.2, 0.5, 0.9\}$) on Qwen3-4B-Base. As shown in Figure 6a, we found that SAE is robust to this hyperparameter, consistently

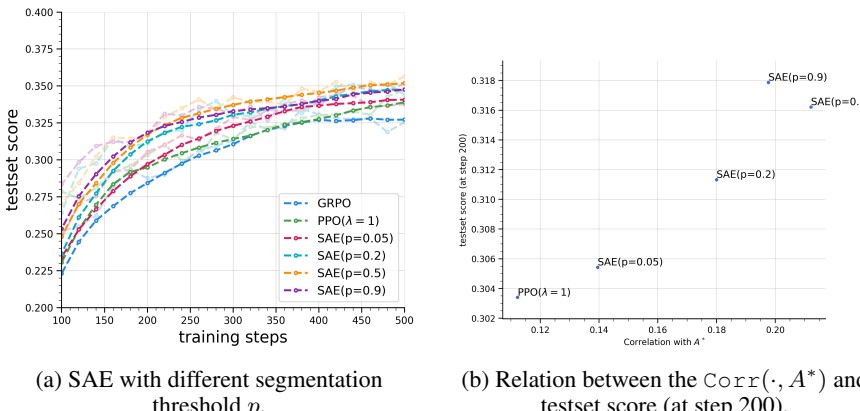

(a) SAE with different segmentation threshold $p$.

(b) Relation between the $\texttt{Corr}(\cdot, A^*)$ and testset score (at step 200).

Figure 6: Ablation on the different segmentation threshold $p$ of SAE. Figure 6a illustrates that SAE consistently outperforms the baseline methods across various values of p. Figure 6b reveals a significant correlation between the performance of each method and the quality of its computed advantage, which is quantified by measuring its correlation with the ground-truth advantage, as defined in Section 5.3.2.

outperforming other baselines for all tested p values. This suggests the performance gains are rooted in the segmental estimation strategy itself, rather than a finely-tuned heuristic.

More importantly, we found that the final task performance is strongly correlated with the quality of the advantage signal, as shown in Figure 6b. It reveals a significant correlation between the performance of each method and the quality of its computed advantage, which is quantified by measuring its correlation with the ground-truth advantage, as defined in Section 5.3.2. This finding provides a principled validation for our probability-based heuristic, directly linking a higher-quality advantage signal to better results. Furthermore, it suggests a promising direction for future work, where $p$ could be dynamically tuned to maximize this correlation, leading to a more adaptive and robust training process.

# 6 CONCLUSION

This paper addresses the challenge of high estimation bias GAE when applying PPO to long-horizon reasoning tasks within the Reinforcement Learning with Verifiable Rewards (RLVR) regime. We introduced Segmental Advantage Estimation (SAE), a novel method that mitigates this bias by replacing GAE's token-level bootstrapping with a more strategic segment-level approach. By partitioning sequences into semantically coherent segments using a probability-based heuristic, SAE selectively computes advantages only at meaningful transition boundaries.

Our empirical evaluation on mathematical problem-solving benchmarks demonstrates that SAE significantly outperforms established baselines, including GRPO and various PPO configurations, in terms of final performance, training stability, and sample efficiency. These improvements are consistent across multiple model scales. Furthermore, a direct correlation analysis validates that our proposed estimator achieves a higher alignment with an approximate ground-truth advantage, substantiating its effectiveness in reducing estimation bias. We also conduct an ablation on SAE's sensitivity to segmentation threshold $p$, showing that SAE is robust to this hyperparameter.

For future work, a primary direction is the exploration of more sophisticated segmentation strategies; our preliminary experiments found that a naive uniform segmentation fails to improve sample efficiency, underscoring the critical role of the segmentation heuristic.

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

# A    SEGMENT ADVANTAGE ESTIMATION: FORMULATION EQUIVALENCE

**Definition 1** (Segment-Aware Discount Function). *Let* $\mathcal{B} = \{t_1, t_2, \ldots, t_K, T\}$ *be a strictly increasing sequence of segment boundaries including the terminal position. For* $t \in \{1, 2, \ldots, T-1\}$ *and* $\lambda \in (0, 1)$*, define:*

$$\lambda_{SAE}(t) = \begin{cases} 1 & if \ (t+1) \notin \mathcal{B} \\ \lambda & if \ (t+1) \in \mathcal{B} \end{cases} \tag{14}$$

**Theorem A.1** (SAE Formulation Equivalence). *The following two SAE formulations are equivalent:*

$$Boundary \ form: A_t^{SAE} = (1 - \lambda) \sum_{k:t_k>t, t_k<T} \lambda^{k-j} \sum_{i=0}^{t_k-t-1} \delta_{t+i} + \lambda^{|\mathcal{B}|-j} \sum_{i=0}^{T-t-1} \delta_{t+i} \tag{15}$$

$$Product \ form: A_t^{SAE} = \sum_{l=0}^{T-t-1} \left( \prod_{i=0}^{l-1} \lambda_{SAE}(t+i) \right) \delta_{t+l} \tag{16}$$

*where* $j = \min\{k : t_k > t, t_k \in \mathcal{B}\}$*.*

*Proof.* We show that each $\delta_{t+l}$ has identical coefficients in both formulations.

In the boundary form, $\delta_{t+l}$ appears in intermediate terms when $t_k \geq t + l + 1$ and in the terminal term. Let $k_l = \min\{k : t_k \geq t + l + 1, t_k < T\}$. The coefficient is:

$$\text{coeff}(\delta_{t+l}) = (1 - \lambda) \sum_{k=k_l}^{|\mathcal{B}|-1} \lambda^{k-j} + \lambda^{|\mathcal{B}|-j} \tag{17}$$

$$= (1 - \lambda)\lambda^{k_l-j} \sum_{m=0}^{|\mathcal{B}|-1-k_l} \lambda^m + \lambda^{|\mathcal{B}|-j} \tag{18}$$

$$= (1 - \lambda)\lambda^{k_l-j} \frac{1 - \lambda^{|\mathcal{B}|-k_l}}{1 - \lambda} + \lambda^{|\mathcal{B}|-j} \tag{19}$$

$$= \lambda^{k_l-j}(1 - \lambda^{|\mathcal{B}|-k_l}) + \lambda^{|\mathcal{B}|-j} = \lambda^{k_l-j} \tag{20}$$

The key observation is that $k_l - j$ equals the number of boundaries crossed from $t$ to $t + l$:

$$k_l - j = \sum_{i=0}^{l-1} \mathbf{1}[(t + i + 1) \in \mathcal{B}] \tag{21}$$

In the product form, the coefficient of $\delta_{t+l}$ is:

$$\prod_{i=0}^{l-1} \lambda_{\text{SAE}}(t + i) = \prod_{i=0}^{l-1} \lambda^{\mathbf{1}[(t+i+1)\in\mathcal{B}]} = \lambda^{\sum_{i=0}^{l-1} \mathbf{1}[(t+i+1)\in\mathcal{B}]} = \lambda^{k_l-j} \tag{22}$$

Therefore, both formulations yield identical coefficients for each $\delta_{t+l}$, establishing equivalence. $\square$

## B  THE UPPER BOUND OF BIAS IN SAE

**Theorem 4.1.** *(Upper bound of bias for* SAE *under uniform segmentation)  Consider the SAE advantage estimation at step $t = 0$ as defined in Eq. 8:*

$$A_0^{SAE} = \sum_{l=0}^{T} \left( \prod_{i=0}^{l-1} \lambda_{SAE}(i) \right) \delta_l, \tag{11}$$

*where $\lambda_{SAE}(t)$ is given by Eq. 9.*

*We make the following assumptions:*

1. *(Uniform segmentation) To facilitate a tractable theoretical analysis, we assume a simplified, uniform segmentation model defined as:*

$$f_s(t) = \begin{cases} 1 & \text{if } t \equiv 0 \pmod{M} \\ 0 & \text{otherwise}. \end{cases} \tag{12}$$

   *This function places boundaries at regular intervals, creating segments of a fixed length $M$. We also assume the segment length $M$ divides the horizon $T$ exactly, i.e., $T = nM$ for some integer $n$.*

2. *(Value function approximation): $V(s_t) = V^*(s_t) + \varepsilon(s_t)$ where $V^*(s_t)$ is the ground-truth value function and $\varepsilon(s_t)$ is the error term. we also assume that $|\varepsilon(s_t)| \leq \alpha \exp\left(\frac{T-t}{\beta}\right)$ with $\alpha > 0, \beta > 0$, which models a common scenario where the value function's approximation error grows for states further away from the end of the trajectory.*

*Then the bias of SAE satisfies:*

$$\left| bias_{A_0^{SAE}} \right| \leq \alpha \exp\left(\frac{T}{\beta}\right) \left[ 1 + \frac{1 - \lambda}{\exp\left(\frac{M}{\beta}\right) - \lambda} \right]. \tag{13}$$

*Consequently, the bias upper bound is inversely related to M: larger values of M yield tighter bounds.*

*Proof.* From the SAE formulation, the bias contribution is:

$$\text{bias}_{A_0^{\text{SAE}}} = \sum_{l=0}^{T-1} \left( \prod_{i=0}^{l-1} \lambda_{\text{SAE}}(i) \right) [\varepsilon(s_{l+1}) - \varepsilon(s_l)] \tag{23}$$

For uniform segmentation, we partition the sum by segments $S_k = [kM, (k+1)M)$ for $k = 0, 1, \ldots, T/M - 1$:

$$\text{bias}_{A_0^{\text{SAE}}} = \sum_{k=0}^{T/M-1} \sum_{l=kM}^{(k+1)M-1} \lambda^k [\varepsilon(s_{l+1}) - \varepsilon(s_l)] \tag{24}$$

$$= \sum_{k=0}^{T/M-1} \lambda^k [\varepsilon(s_{(k+1)M}) - \varepsilon(s_{kM})] \tag{25}$$

where we used the telescoping property within each segment and $\prod_{i=0}^{kM-1} \lambda_{\text{SAE}}(i) = \lambda^k$.

Expanding the telescoping sum:

$$\text{bias}_{A_0^{\text{SAE}}} = [\varepsilon(s_M) - \varepsilon(s_0)] + \lambda[\varepsilon(s_{2M}) - \varepsilon(s_M)] \tag{26}$$

$$+ \lambda^2[\varepsilon(s_{3M}) - \varepsilon(s_{2M})] + \ldots + \lambda^{T/M-1}[\varepsilon(s_T) - \varepsilon(s_{(T/M-1)M})] \tag{27}$$

Collecting terms by $\varepsilon(s_{kM})$ coefficients:

$$\text{bias}_{A_0^{\text{SAE}}} = -\varepsilon(s_0) + \varepsilon(s_M)(1-\lambda) + \varepsilon(s_{2M})\lambda(1-\lambda) \tag{28}$$

$$+ \varepsilon(s_{3M})\lambda^2(1-\lambda) + \ldots + \varepsilon(s_{(T/M-1)M})\lambda^{T/M-2}(1-\lambda) + \lambda^{T/M-1}\varepsilon(s_T) \tag{29}$$

Under the assumption $\varepsilon(s_T) = 0$, we obtain:

$$\text{bias}_{A_0^{\text{SAE}}} = -\varepsilon(s_0) + \sum_{k=1}^{T/M-1} \lambda^{k-1}(1-\lambda)\varepsilon(s_{kM}) \tag{30}$$

Applying the triangle inequality and error bound assumption:

$$\left| \text{bias}_{A_0^{\text{SAE}}} \right| \leq |\varepsilon(s_0)| + \sum_{k=1}^{T/M-1} \lambda^{k-1}(1-\lambda)|\varepsilon(s_{kM})| \tag{31}$$

$$\leq \alpha \exp\left(\frac{T}{\beta}\right) + (1-\lambda) \sum_{k=1}^{T/M-1} \lambda^{k-1}\alpha \exp\left(\frac{T-kM}{\beta}\right) \tag{32}$$

$$= \alpha \exp\left(\frac{T}{\beta}\right) \left[ 1 + (1-\lambda) \sum_{k=1}^{T/M-1} \lambda^{k-1} \exp\left(\frac{-kM}{\beta}\right) \right] \tag{33}$$

For the geometric series with $\mu = \lambda \exp\left(\frac{-M}{\beta}\right) < 1$:

$$\sum_{k=1}^{T/M-1} \lambda^{k-1} \exp\left(\frac{-kM}{\beta}\right) = \lambda^{-1} \sum_{k=1}^{T/M-1} \mu^k \leq \sum_{k=1}^{\infty} \mu^k = \frac{\mu}{\lambda(1-\mu)} = \frac{\exp\left(\frac{-M}{\beta}\right)}{1 - \lambda \exp\left(\frac{-M}{\beta}\right)} \tag{34}$$

Therefore:

$$\left| \text{bias}_{A_0^{\text{SAE}}} \right| \leq \alpha \exp\left(\frac{T}{\beta}\right) \left[ 1 + \frac{(1-\lambda)\exp\left(\frac{-M}{\beta}\right)}{1 - \lambda \exp\left(\frac{-M}{\beta}\right)} \right] = \alpha \exp\left(\frac{T}{\beta}\right) \left[ 1 + \frac{1-\lambda}{\exp\left(\frac{M}{\beta}\right) - \lambda} \right] \tag{35}$$

$\square$

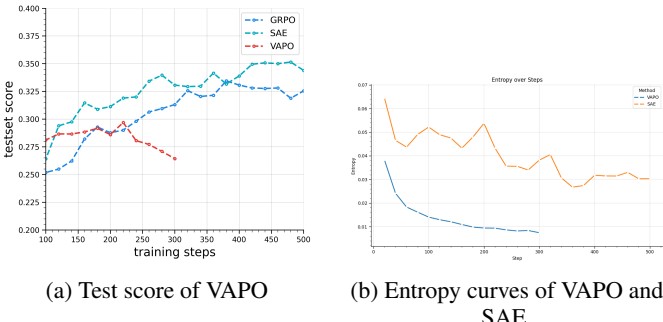

(a) Test score of VAPO      (b) Entropy curves of VAPO and
SAE.

Figure 7: Comparison of re-implemented VAPO and SAE. (left) The testscore growth of VAPO plateaued after approximately 200 steps, (right) possibly due to the fast diminishment of entropy loss.

## C  THE USE OF LARGE LANGUAGE MODELS

We employ LLMs to help writing this paper, but only for the purpose of polishing writing. Specifically, we use LLMs to translate or rewrite specific sentences to improve the flow of the article. Each sentence generated by LLM is carefully checked.

## D  TRAINING DETAILS OF SECTION 5.3.1

For the ablation study on model size, we adopted the same configuration as described in Section 5.1, with the sole modification being the initial model size across different experimental runs.

For the domain-specific experiments, a universal model size of 4B was employed due to computational resource constraints. In comparison to the setup in Section 5.1, the primary distinctions reside in the model and the data, as detailed in the table below:

**Domain: CODE**

- **Initial Model:** Qwen3-4B-Instruct-2707
- **Training Data:** AReal boba2 (inclusionAI, 2025; Luo et al., 2025)
- **Test Data:** APPS(Hendrycks et al., 2021), Codecontests(Li et al., 2022), Codeforces(MatrixStudio, 2024), and TACO(Li et al., 2023)

**Domain: STEM**

- **Initial Model:** Qwen3-4B-Base
- **Training Data:** SCP-116K-cleaned (Lu et al., 2025), adopted form ProRL(Mingjie Liu, 2025)
- **Test Data:** GPQA-Diamond(Rein et al., 2024)

## E  REIMPLEMENTATION OF VAPO

VAPO is one of our baseline models. However, its implementation is not open-source, and there is no publicly available code that can reproduce the results reported in the original paper. Consequently, in our preliminary experiments, we endeavored to implement VAPO, incorporating its key features such as the positive-example LM loss and adaptive $\lambda$. We then conducted experiments on the Qwen3-4B model using the VAPO training configuration. As illustrated in Figure 7a, our VAPO implementation exhibited rapid initial performance gains, but its growth plateaued after approximately 200 steps. Further analysis, shown in Figure 7b, revealed that the entropy loss diminished

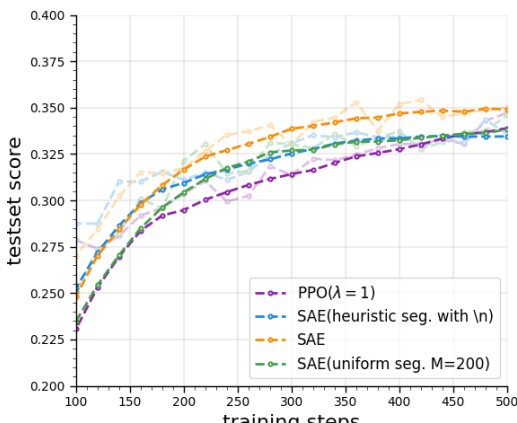

Figure 8: SAE variants with different segmentation methods. SAE's probability-based segmentation methods achieves the best performance.

too quickly. In reinforcement learning, a rapid decrease in entropy can lead to suboptimal performance by prematurely reducing policy exploration. This premature convergence on a specific strategy can prevent the agent from discovering a more optimal policy.

Therefore, to create a more effective and controlled baseline, we isolated the most critical component of VAPO relevant to our work: the use of an adaptive lambda for advantage calculation. All other experimental variables were kept identical to those of our proposed method. This approach established a more rigorous baseline and ultimately yielded superior results compared to the original VAPO configuration.

## F    DIFFERENT SEGMENTATION METHODS

A key innovation of SAE is its segmentation method, which is based on the model's intrinsic uncertainty by leveraging features from its output tokens. To isolate and evaluate the effectiveness of this specific component, we conducted an ablation study on the Qwen3-4B-base model. This study compared two alternative segmentation strategies against SAE's proposed method: (1)Uniform Segmentation: The sequence was divided into fixed-length segments of M=200. (2) Newline Character Segmentation: The sequence was segmented based on the occurrence of the newline character $\backslash n$. For SAE, we set $p = 0.5$.

The experimental results are presented in Figure 8. Notably, all variants of SAE demonstrated a faster increase in test scores compared to the PPO ($\lambda$=1) baseline. This finding underscores the fundamental importance of the segmentation introduced by SAE. Furthermore, a direct comparison of the segmentation methods reveals that SAE's probability-based segmentation outperforms the two other variants. This indicates that leveraging the model's own uncertainty for segmentation is a more effective strategy.

## G    AVERAGE SEGMENTATION LENGTH FOR DIFFERENT SEGMENTATION THRESHOLD P

In Figure 9, we visualize the average segmentation length for different $p$ (defined in Eq 5). We can see that bigger model generally yields smaller average segmentation length for a given $p$, and the segmentation length grows fastly when $p \to 0$.

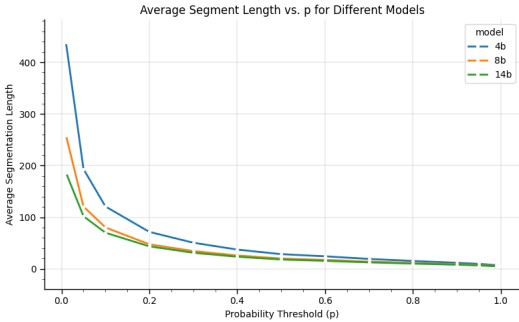

Figure 9: Average segmentation length for different segmentation threshold $p$. Bigger model yields smaller average segmentation length for a given $p$.

