# OpenReview forum: "Segmental Advantage Estimation: Enhancing PPO for Long-Context LLM Training"
_ICLR.cc/2026/Conference — Submitted to ICLR 2026_

### Official Review · Reviewer_yZzM · 2025-11-01

**Soundness:** 4
**Presentation:** 4
**Contribution:** 3
**Rating:** 8
**Confidence:** 4

**Summary:**

The paper focuses on the problem of instability of GAE estimation of token-level PPO method, when used in a typical RLVR setting with terminal reward. Most of the contemporary solutions employ $\lambda=1$ further amplifying the instability in training an accurate value estimate.

To mitigate this issue, authors propose a clever solution of segmenting the responses into chunks of high probablity tokens and reducing the overall effective "actions" in a response trajectory from MDP perspective. They call this method Segmental Advantage Estimation (SAE). SAE effectively reduced the number of steps in a response trajectory from number of tokens to number of segmented chunks and in-effect reduces the inefficiency in estimating the advantage estimate.

Experiments on multiple model scales across standard math datasets and evals, show that SAE consistently outperforms PPO ($\lambda=1$ and adaptive $\lambda$) and GRPO.

**Strengths:**

- Well motivated problem of instablity of GAE estimation in RLVR where $\lambda$ is set to 1
- Insightful solution to focus on segments of response for GAE estimation instead of per token
- Theoretical analysis to justify that SAE reduces the bias in estimation.
- Emprirical analysis showcasing SAE has highest correlation with true Advantage compared to other baselines in a controlled setting.

**Weaknesses:**

- The SAE method uses a fixed threshold of 0.2 on the probability to decide the segments. I would have preferred an abilation study for the choice of this parameter.
- I would prefer to have SAE compared with the simple baseline of fixed length segments from the theoretical analysis of section 4.2. For example, what is the effect when I naive let chunks to be of size $M=100$ or $200$ tokens irrespective of the probablity. Does the choice of segmentation method matter towards the downstream performance of SAE?

**Questions:**

1. I would have preferred to see some analysis of average segment length for different model sizes in a practical setting. The paper will greatly benefit with more analysis and the effects of varying the probablity threshold $p$ on the segment size.
2. Did authors try other segmentation methods such as entropy instead of raw token probability?

---

> ### Author Response · Authors · 2025-11-24
> **Response to Reviewer**
>
> We are truly grateful for your thorough review and your exceptionally positive assessment of our work. We are delighted that you found our problem motivation clear, our solution insightful, and our analyses compelling. Your suggestions for further ablation and analysis are excellent, and we believe they significantly strengthen the paper.
>
> We are happy to report that we have incorporated all of your suggestions into the revised manuscript. Below, we detail how we have addressed each of your points.
>
> > Weakness 1: The SAE method uses a fixed threshold of 0.2 on the probability to decide the segments. I would have preferred an ablation study for the choice of this parameter.
>
> We completely agree that an ablation study on the hyperparameter p is crucial for understanding the robustness of SAE. As you suggested, we have added a comprehensive ablation study in Section 5.3.3 of our revised paper. This new section demonstrates two key findings:
> - Robustness to p: SAE consistently outperforms all baselines across a wide range of p values (0.05, 0.2, 0.5, and 0.9), confirming that our method is not overly sensitive to this choice.
> - A Principled Metric for Tuning p: We discovered a strong correlation between final task performance and the quality of the advantage signal (measured by its correlation with the ground-truth advantage, A*). This provides a principled, empirical basis for selecting an optimal p in practice.
>
> > Weakness 2: I would prefer to have SAE compared with the simple baseline of fixed length segments... Does the choice of segmentation method matter towards the downstream performance of SAE?
>
> This is an excellent question that directly tests the novelty and contribution of our probability-based segmentation heuristic. To address this, we have added a new set of experiments in Appendix F.
> In this section, we compare our proposed method against SAE variants that use simpler heuristics, including uniform (fixed-length) segmentation and segmentation by newline characters (\n). The results show that:
> - Segmentation Matters: The choice of segmentation method does significantly impact downstream performance, with our probability-based heuristic achieving the best results.
> - The Core Idea is Robust: Crucially, even the SAE variants with naive segmentation heuristics still outperform the token-level PPO baselines. This provides strong evidence for our core thesis: the primary benefit comes from the segmental estimation strategy itself, which effectively reduces estimation bias.
>
> > Questions (1): I would have preferred to see some analysis of average segment length... and the effects of varying the probability threshold p on the segment size.
>
> To provide a clearer picture of the effect of p, we have included a new visualization in Appendix G. This figure illustrates how the average segment length varies as a function of the probability threshold p, directly addressing your question.
>
>
> > Question(2): Did authors try other segmentation methods such as entropy instead of raw token probability?
>
> This is a great suggestion. We did not explicitly experiment with entropy. However, token entropy (or surprisal) is conceptually very similar to low token probability, as both are designed to capture moments of high uncertainty in the generation process. Given the strong performance and simplicity of the probability-based heuristic, we believe it serves as a good and intuitive starting point. Exploring entropy and other more sophisticated metrics is a very promising direction for future work.

---

### Official Review · Reviewer_ikv3 · 2025-11-01

**Soundness:** 2
**Presentation:** 3
**Contribution:** 2
**Rating:** 2
**Confidence:** 3

**Summary:**

The paper proposes the Segmental Advantage Estimation (SAE) method to improve value estimation in PPO algorithms. SAE first partitions the generated sequence into coherent sub-segments, using low-probability tokens as heuristic boundaries, and then treats each segment as an action for GAE computation. Experiments are conducted to demonstrate the effectiveness of the proposed method compared to standard PPO.

**Strengths:**

- The paper is well written and easy to follow.
- Accurate value estimation is crucial for PPO algorithms. The idea of segmenting responses based on low-probability tokens is intuitive and makes sense.
- Experiments are conducted to demonstrate the effectiveness of the proposed method compared to standard PPO.

**Weaknesses:**

My main concern is the lack of comparison with related baseline:
-  There is no comparison with the mentioned related works, such as VC-PPO and VAPO
-  Previous studies have proposed computing GAE at the step level (e.g., by splitting sequences using special tokens such as ‘\n’) [1]. This paper is closely related to those approaches, and a comparison with them would help better demonstrate the effectiveness of the proposed method.

[1]Chen, Guoxin, et al. "Alphamath almost zero: process supervision without process." Advances in Neural Information Processing Systems 37 (2024): 27689-27724

**Questions:**

Please refer to the weakness part.

---

> ### Author Response · Authors · 2025-11-24
> **Response to Reviewer**
>
> We sincerely thank you for your review and for your positive feedback on the clarity of our writing and the intuitiveness of our core idea. Your main concern revolves around the lack of comparison with specific related works, namely VAPO and step-wise methods like those in AlphaMath. We believe there may have been a misunderstanding, as we did conduct the comparison with VAPO and ensure our experimental setup was aligned with best practices from the field.
>
>
>
> Below, we address your concerns point by point.
>
>
> > Weakness 1: There is no comparison with the mentioned related works, such as VC-PPO and VAPO.
>
> This is a critical point, and we apologize if our presentation was not clear.
>
> - Comparison with VAPO: We benchmark against VAPO in two ways:
>     - In the Main Paper (original version): The baseline labeled "PPO (adaptive λ)" (Table 1, Figure 2) directly implements the core GAE-related innovation from the VAPO paper—an adaptive lambda strategy based on response length. To ensure a fair and focused comparison of advantage estimation methods, we isolated this key component.
>     - In the Appendix (revised version): For completeness, we have now implemented the full VAPO system, including all its orthogonal components (e.g., the positive-sample LM loss). This head-to-head comparison is presented in Appendix E of the revised paper. The results clearly show that SAE still holds a significant performance advantage over the full VAPO implementation.
> - Relation to VC-PPO: VC-PPO does not introduce a novel advantage estimation method; instead, it focuses on stabilizing value function training. Its techniques are therefore orthogonal to SAE. In fact, following best practices, we have incorporated key techniques from VC-PPO, such as value pre-training, into all PPO-based methods in our study, including our own. This ensures our baselines are strong and the comparison is fair.
>
> We trust this clarifies that we have not only compared against VAPO but have also integrated insights from VC-PPO to strengthen our entire experimental framework.
>
>
> > Weakness 2: Previous studies have proposed computing GAE at the step level (e.g., by splitting sequences using special tokens such as ‘\n’). This paper is closely related to those approaches...
>
> We thank you for pointing out this line of work, and we have involved it in our related works. We agree that step-wise breakdown is a general idea, but we would like to respectfully clarify the fundamental distinction between our work (SAE) and search-based methods like AlphaMath:
>
>
> - Different Problems and Goals:
>     - SAE is an advantage estimation algorithm designed to improve credit assignment for a given trajectory within a standard PPO framework.
>     - AlphaMath is a search-based sampling algorithm that uses MCTS to generate high-quality positive trajectories. It does not perform GAE computation in the traditional sense and lacks core concepts like the $\lambda$ hyperparameter, which manages the bias-variance trade-off.
>     - Therefore, they are not direct competitors but rather orthogonal approaches: AlphaMath focuses on improving data quality for RL, while SAE focuses on improving the RL algorithm's learning from that data.
>
>
> - Novelty and Superiority of Our Segmentation Method: A key novelty of our work lies in proposing a more effective, probability-based segmentation method. We empirically validated its superiority over simpler heuristics like splitting by newline characters (\n):
>     - Correlation with Ground-Truth: Our analysis shows that \n-based segmentation yields an advantage signal with a Pearson correlation of only 0.12 to the approximate ground-truth advantage (A*). In stark contrast, SAE's method achieves a much higher correlation of 0.203, indicating a significantly more accurate signal.
>     - End-to-End Performance: In Appendix F, we provide end-to-end training results that directly compare different segmentation heuristics. These experiments confirm that our probability-based approach leads to substantially better final performance than \n-based segmentation.
>
>
> In summary, not only is AlphaMath conceptually distinct, but our core technical contribution in segmentation is novel and demonstrably more effective than related heuristics. We hope these clarifications fully address your concerns and highlight the contributions of our work.

---

### Official Review · Reviewer_BbQN · 2025-11-02

**Soundness:** 2
**Presentation:** 2
**Contribution:** 2
**Rating:** 4
**Confidence:** 3

**Summary:**

This paper introduces Segmental Advantage Estimation (SAE) to improve Proximal Policy Optimization (PPO) for training Large Language Models (LLMs) on long-horizon reasoning tasks with verifiable rewards (RLVR).  It aims to address the unreliable advantage estimation in sparse-reward settings, where traditional Generalized Advantage Estimation (GAE) amplifies bias by performing token-level bootstrapping using noisy value predictions. SAE mitigates this by first partitioning the generated sequence into semantically coherent segments, using low-probability tokens as heuristic boundaries, and then selectively computing advantages only at these segment transitions. This reduces bootstrapping bias by filtering out noise from intermediate, low-information tokens.

**Strengths:**

(1) The proposed method in this paper is practically elegant, as its recursive formulation allows for seamless integration into existing PPO frameworks with minimal computational overhead.
(2) The empirical evaluation is thorough, benchmarking against strong baselines like GRPO and adaptive PPO variants across multiple out-of-distribution test sets (AIME, AMC). The consistent performance gains across 4B, 8B, and 14B model sizes strongly support the method's robustness and scalability.

**Weaknesses:**

(1) While the use of low-probability tokens is intuitive, it is an unsupervised method that may not always align perfectly with true semantic boundaries, potentially introducing its own form of noise.
(2) The evaluation is confined to mathematical reasoning. While this is a canonical domain for RLVR, the paper does not demonstrate SAE's efficacy in other long-context scenarios like code generation or complex dialogue, limiting the claimed generality of the approach.

**Questions:**

(1) How might more sophisticated, learned segmentation strategies (e.g., leveraging an auxiliary model or syntactic features) further improve the performance and robustness of SAE compared to the current probability-based heuristic?
(2) The paper sets the segmentation threshold p=0.2 universally. How sensitive is the performance of SAE to this hyperparameter, and could an adaptive or dynamically learned threshold offer benefits?

---

> ### Author Response · Authors · 2025-11-24
> **Response To the Reviewer (Part1)**
>
> We sincerely thank you for your thoughtful and constructive review of our paper. We are delighted that you recognized the practical elegance, seamless integration, and strong empirical performance of our proposed SAE method. Your feedback is valuable, and we appreciate the opportunity to address your concerns and further clarify our contributions.
>
> In summary, we have provided new experimental evidence demonstrating that:
> (a) The core benefit of SAE stems from the **segmental estimation strategy itself**, which is robust even to sub-optimal heuristics (e.g., uniform segmentation).
> (b) Our probability-based segmentation heuristic is **robust to its hyperparameter** and can be adaptively tuned guided by a principled metric.
> (c) The benefits of SAE generalize beyond mathematical reasoning to other challenging long-horizon domains, including **CODE and STEM**.
>
> We believe these additions, which we have incorporated into the revised version of the paper, substantially strengthen our contribution. We thank you once again for your time and valuable feedback.

---

> > ### Author Response · Authors · 2025-11-24
> > **Response To the Reviewer (Part2)**
> >
> > Below, we address the weaknesses and questions you raised term-by-term.
> >
> > ---
> >
> > > **Weakness (1)** - While the use of low-probability tokens is intuitive, it is an unsupervised method that may not always align perfectly with true semantic boundaries, potentially introducing its own form of noise.
> >
> > We agree that our probability-based heuristic is unsupervised and may not achieve perfect alignment with true semantic boundaries. However, our new experimental results demonstrate that the core segmental estimation strategy is remarkably robust to such potential mismatches, based on the following two findings:
> >
> > *   **[Figure 6(a) in Section 5.3.3] SAE is robust to the threshold `p`:** Our new experiments show that SAE's performance is not overly sensitive to the choice of `p`. We found that SAE consistently outperforms all baselines across a wide range of `p` values (from 0.05 to 0.9).
> >
> > *   **[Figure 8 in Appendix F] Alternative segmentation methods also prove effective:** We tested several SAE variants using different segmentation methods. Even simpler heuristics, such as uniform segmentation or splitting by newline characters (`\n`), deliver superior performance compared to the baselines, although they do not reach the full potential of our proposed method.
> >
> > These results strongly suggest that the primary benefit arises from reducing the frequency of noisy bootstrapping, rather than from a perfectly-tuned heuristic. Therefore, the risk of "introducing its own form of noise" from potential mismatches is minimal, as the segmental approach provides a strong performance floor well above the baselines.
> >
> > ---
> >
> > > **Weakness (2)** - The evaluation is confined to mathematical reasoning. While this is a canonical domain for RLVR, the paper does not demonstrate SAE's efficacy in other long-context scenarios like code generation or complex dialogue, limiting the claimed generality of the approach.
> >
> > This is an excellent point. While mathematical reasoning is a canonical benchmark, we agree that demonstrating broader applicability is crucial. Accordingly, we have expanded our evaluation to two additional long-horizon domains: **CODE and STEM**. The results are presented in the revised paper (Figure 4, Section 5.3.1).
> >
> > The results show that SAE's benefits generalize effectively. Notably, in the code generation domain, SAE achieves a remarkable **~12 percentage point** lead over GRPO after 500 training steps. These cross-domain results, combined with our multi-model-size experiments, provide strong evidence that SAE is indeed a more general technique for improving RL in long-context tasks.

---

> > > ### Author Response · Authors · 2025-11-24
> > > **Response To the Reviewer (Part3)**
> > >
> > > > **Question (1):** How might more sophisticated, learned segmentation strategies (e.g., leveraging an auxiliary model or syntactic features) further improve the performance and robustness of SAE compared to the current probability-based heuristic?
> > >
> > > This is a very insightful question. As our new ablation study in Figure 8 (Appendix F) illustrates, segmentation quality does influence final performance. We therefore agree that more sophisticated, learned segmentation strategies could potentially offer more precise boundary detection and yield further improvements.
> > >
> > > However, our choice of the probability-based heuristic was a deliberate design decision prioritizing **simplicity, generality, and practical applicability**. As you kindly noted in the strengths, one of SAE's key advantages is its elegance and minimal computational overhead. Our method leverages the model's intrinsic signals without requiring auxiliary models, complex feature engineering, or disruptive changes to the training pipeline. This ensures SAE is exceptionally easy to adopt and integrate into existing PPO frameworks. While learned segmenters are a very promising avenue for future research, our current method establishes a strong and practical baseline that already delivers significant gains.
> > >
> > > ---
> > >
> > > > **Question (2):** The paper sets the segmentation threshold p=0.2 universally. How sensitive is the performance of SAE to this hyperparameter, and could an adaptive or dynamically learned threshold offer benefits?
> > >
> > > As detailed in our response to Weakness (1) and shown in Figure 6(a), our new experiments confirm that **SAE is robust to the choice of `p` across a wide range** (0.05 to 0.9).
> > >
> > > More importantly, our investigation revealed a principled approach for guiding the choice of `p` and opened a clear path toward an adaptive method. As shown in **Figure 6(b)**, we found a strong positive correlation between final task performance and the quality of the advantage signal (quantified by its correlation with the ground-truth advantage, `A*`, from Section 5.3.2).
> > >
> > > This finding provides direct validation for our method—it confirms that a better advantage signal leads to better results. It also suggests **a concrete mechanism for a dynamic threshold: one could periodically estimate `A*` during training and select the `p` that maximizes this correlation**. While we leave the implementation of such an adaptive system as future work to maintain the simplicity of the current SAE framework, this finding firmly grounds our heuristic in a measurable, principled metric.

---

### Author Response · Authors · 2025-11-24
**Summary of Revision**

To enhance the robustness and comprehensiveness of our paper, we have incorporated several new experiments and analyses based on the reviewers' valuable feedback. The revised content is highlighted with blue in the revised manuscript. The key additions of the revised paper are summarized below:

- (Section 5.3.3) Sensitivity to Hyperparameter p (with p = 0.05, 0.2, 0.5, 0.9)
- (Section 5.3.1) Generalization to New Domains (CODE & STEM)
- (Appendix E) Full System Comparison with VAPO
- (Appendix F) Comparison of different Segmentation Methods(uniform segmentation & \n-segmentation)
- (Appendix G) Impact of p on Segment Length

---

### Author Response · Authors · 2025-12-02
**Summary of Rebuttal**

To the Area Chair,

Recognizing the significant workload for ACs during this ICLR cycle, and to help ease the review process, we have prepared this brief summary of the reviewers' concerns and our corresponding rebuttals and revisions. We sincerely thank the reviewers for their insightful feedback, which has significantly strengthened our paper. We have submitted a revised manuscript incorporating new experiments and analyses that directly address their concerns.

**Reviewer BbQN (Rating 4):**

*   **Concern 1: Robustness of the segmentation heuristic.** The reviewer questioned if the probability-based heuristic might be noisy.
    *   **Our Rebuttal & Revision:** We conducted new experiments (**Sec 5.3.3, Appendix F**) demonstrating that:
        1.  SAE is robust to its hyperparameter `p` across a wide range of values.
        2.  The core benefit stems from the **segmental approach itself**, as even simpler heuristics (e.g., uniform, newline segmentation) significantly outperform token-level baselines.

*   **Concern 2: Limited evaluation scope (only mathematical reasoning).**
    *   **Our Rebuttal & Revision:** We have expanded our evaluation to two additional long-horizon domains: **CODE and STEM (Sec 5.3.1)**. The results confirm that SAE's benefits generalize, with a particularly strong performance uplift in code generation.

*   **Question: Potential for more sophisticated, adaptive segmentation.** The reviewer asked about learned segmentation strategies.
    *   **Our Rebuttal & Revision:** We explained that our choice prioritizes simplicity and practicality, a strength the reviewer acknowledged. Crucially, our new analysis in **Sec 5.3.3 (Figure 6b)** provides a principled path toward an adaptive method by showing a **strong correlation between our advantage signal's quality and final task performance**. This grounds our heuristic in a measurable metric that could guide a dynamic threshold.

**Reviewer ikv3 (Rating 2):**

*   **Concern 1: Lack of comparison with VAPO.**
    *   **Our Rebuttal & Revision:** We clarified that our **"PPO(adaptive λ)" baseline already implemented VAPO's core advantage estimation mechanism**. For completeness, we have now added a **full head-to-head comparison with VAPO in Appendix E**, where SAE demonstrates superior performance.

*   **Concern 2: Perceived lack of novelty compared to MCTS-based samplers (e.g., AlphaMath)**
    *   **Our Rebuttal & Revision:** We clarified the conceptual distinction: SAE is a **PPO advantage estimation algorithm**, while AlphaMath is a **search-based MCTS sampler** focused on data generation, which are fundamentally different from PPO/GAE. They are orthogonal. More importantly, we provided new experiments in **Appendix F** directly comparing our probability-based segmentation against simpler heuristics (like `\n`-based segmentation), empirically proving that **our proposed method is novel and leads to substantially better performance**.

**Reviewer yZzM (Rating 8):**

*   **Suggestions:** The reviewer, while very positive, suggested further analysis to strengthen the paper, including an ablation on the hyperparameter `p`, a comparison with fixed-length segmentation, and an analysis of segment length.
    *   **Our Rebuttal & Revision:** We are grateful for these excellent suggestions and have incorporated all of them into the revised paper:
        1.  A comprehensive **ablation study on the hyperparameter `p`** (Sec 5.3.3).
        2.  A comparison with **simpler segmentation heuristics**, including uniform (fixed-length) segmentation (Appendix F).
        3.  An analysis of the **impact of `p` on average segment length** (Appendix G).

We believe these additions substantially bolster our claims regarding SAE's robustness, generality, and novelty. We are confident that the revised paper now presents a more complete and compelling contribution.

---

### Meta-Review · Area_Chair_xUZY · 2026-01-01

**Summary:**

This paper proposes to partition generated sequence into coherent sub-segments using low-probability tokens as boundaries, and selectively compute advantages only at segment transitions to alleviate the bias of GAE that relies on token-level bootstrapping for value predictions. Experiments across various model scales are conducted to compare the performance of the proposed method with PPO and GRPO.

The paper receives diverse evaluations. The major concerns of reviewers include:

1) sensitivity of the threshold hyperparameter required by the method, as raised by Reviewers BbQN and yZzM

2) the uncertain optimality of the proposed partition strategy and its novelty and superiority compared with other simple alternatives e.g. splitting using tokens ‘\n’, as raised by Reviewers BbQN, ikv3, and yZzM

3) limited evaluation scope and lack of comparison with more baselines, as raised by Reviewers BbQN and ikv3

**Reviewer Concerns:**

After reading the paper and all the comments, the AC thinks that the major concern (2) is still outstanding. The technical novelty of partitioning sequences by low-probability tokens is minor. As Reviewer ikv3 suggested, prior studies have already adopted step-level advantage estimation by other splitting choices. The authors added comparison with these alternative splitting strategies in the rebuttal, but the results show that the proposed method only surpasses them by a small margin. Therefore, this issue that undermines the quality and significance of this work has not been resolved.

**Reviewer Scores:**

The final scores would be 2, 4, 8.

---

### Decision · Program_Chairs · 2026-01-26

Reject